# Molecular Imaging of Vulnerable Coronary Plaque with Radiolabeled Somatostatin Receptors (SSTR)

**DOI:** 10.3390/jcm10235515

**Published:** 2021-11-25

**Authors:** Luz Kelly Anzola, Jose Nelson Rivera, Juan Carlos Ramirez, Alberto Signore, Fernando Mut

**Affiliations:** 1Nuclear Medicine Department, Clinica Reina Sofia, Bogota 110121, Colombia; 2Nuclear Medicine Department, Clinica Universitaria Colombia, Bogota 111021, Colombia; 3Postgraduate Program in Nuclear Medicine Fundacion, Universitaria Sanitas, Bogota 111021, Colombia; jc.ramirezye@unisanitas.edu.co; 4Internal Medicine Department, Clinica Reina Sofia, Bogota 110121, Colombia; jonerimo@yahoo.com; 5Nuclear Medicine Unit, Department of Medical-Surgical Sciences and Translational Medicine, Faculty of Medicine and Psychology, “Sapienza” University, 00189 Rome, Italy; alberto.signore@uniroma1.it; 6Nuclear Medicine Department, Hospital Italiano, Montevideo 11600, Uruguay; mut.fer@gmail.com

**Keywords:** atherosclerosis, vascular inflammation, PET/CT, ^68^Ga-DOTA-TATE, activated macrophages

## Abstract

Atherosclerosis is responsible for the majority of heart attacks and is characterized by several modifications of the arterial wall including an inflammatory reaction. The silent course of atherosclerosis has made it necessary to develop predictors of disease complications before symptomatic lesions occur. Vulnerable to rupture atherosclerotic plaques are the target for molecular imaging. To this aim, different radiopharmaceuticals for PET/CT have emerged for the identification of high-risk plaques, with high specificity for the identification of the cellular components and pathophysiological status of plaques. By targeting specific receptors on activated macrophages in high-risk plaques, radiolabelled somatostatin analogues such as ^68^Ga-DOTA-TOC, TATE,0 or NOC have shown high relevance to detect vulnerable, atherosclerotic plaques. This PET radiopharmaceutical has been tested in several pre-clinical and clinical studies, as reviewed here, showing an important correlation with other risk factors.

## 1. Introduction

Cardiovascular atherosclerotic disease is a systemic condition that affects major arteries of the body, being the most common cause of death in the western world and expected to reach 22.2 million people by 2030 [1]. Sudden death is the first manifestation of the disease in a large proportion of patients [2]. As atherosclerotic plaque rupture is the underlying cause in the majority of patients suffering hard events, the ultimate goal for imaging in cardiovascular disease should be a non-invasive approach capable of identifying vulnerable plaques [3]. The physiopathology of atherosclerosis is characterized by the presence of complex events, where an inflammatory cascade is triggered by the deposit of low-density lipoproteins (LDL) at sites of endothelial injury, and by the posterior recruiting of macrophages, which catch the oxidized LDL remnants [4]. Advances in the understanding of the pathophysiology of atherosclerosis has allowed for the development of novel diagnostic strategies and provided important insight into the molecular pathways of plaque rupture. However, a screening method to identify patients at risk of acute coronary events is still lacking due to a limited insight into the pathophysiology of the vulnerable plaque and a lack of cost-effective interventions [5].

Considering the dynamic nature of coronary atherosclerotic lesions, it is not easy to identify which plaques will become unstable based on just a single imaging modality. An approach that allows for the identification of cap thickness, necrotic core areas, and the degree of inflammation of the plaque is of paramount clinical relevance. Given the involved costs, the time required, and the potential adverse effects, invasive techniques are not suitable for screening asymptomatic patients. CT angiography can be used to identify the presence of lesions with high-risk characteristics, i.e., those with areas of low attenuation and expansive remodeling. Serial CT could also determine dynamic changes in the plaque as an indicator of instability, warning the clinician to adopt prophylactic measures [6]. In this scenario, it is also important to consider the group of asymptomatic patients with small, angiographically invisible plaques with a potential to rupture due to inflammatory changes that could precipitate a hard event; in such cases, metabolic images reveal evidence of the synergistic role of both imaging modalities.

Non-invasive imaging of atherosclerotic plaques includes optical imaging, computed tomography (CT), magnetic resonance imaging (MRI), ultrasound (US), single-photon emission computed tomography (SPECT), and positron emission tomography (PET). Furthermore, hybrid PET systems coupled with CT or MRI are available, providing both anatomical and molecular information about the disease [7,8]. PET imaging is based on the use of selective radiopharmaceuticals for targeting specific biochemical processes in vivo [9], such as the cellular density of certain receptors or the metabolic activity of a plaque, providing information at the cellular and sub-cellular level for detecting high-risk atherosclerotic lesions [10]. However, limited spatial resolution becomes the most important weakness for evaluating atherosclerotic plaques; for overcoming this limitation, hybrid systems have emerged as state-of-the-art technology for simultaneous morphological and functional imaging [11,12]. Co-registration of PET images with CT or MRI gives a precise anatomical localization of the PET signal and since uptake quantification by SUV may not reflect the true tracer uptake within the lesion, the concomitant use of contrast-enhanced CT or MRI could help to mitigate this source of error. Co-registered CT images permit attenuation correction, which is critical in the diagnostic approach of small structures. Co-registered MRI images also provide soft tissue contrast information and the possibility to evaluate the presence of thrombus. As the quality of PET images could be affected by cardiac and respiratory motion, different technical strategies should be applied, such as the use of end-diastolic images [13] and software-based motion correction techniques, such as motion tracking with 4D CT and data-drive correction respiratory gating from 4D PET [4,5], improving the detection of the small foci of uptake within the coronary arteries. High-resolution instruments have been recently introduced with the possibility to image lesions as small as 1–2 mm in size, including the new CZT cameras [14] and the total body digital PET [15]. Finally, as part of the strategies to improve the detection of the small foci of uptake in atherosclerotic plaques, the development of target-specific radiotracers has resolved the issue of interfering myocardial uptake and the necessity of a faster clearance of the tracer from the blood stream.

^18^F-fluoro-deoxyglucose (^18^F-FDG) is the most widely used and validated PET radiopharmaceutical for atherosclerosis imaging, since plaque uptake is an indicator of increased macrophage activity. Indeed, ex vivo studies have shown a strong correlation between ^18^F-FDG uptake and both macrophage density and the increased gene expression of inflammation markers [16]. Rudd et al. [17] reported one of the first clinical studies of atherosclerosis imaging with ^18^F-FDG PET in patients with transient ischemic attacks, observing significant radiopharmaceutical uptake in symptomatic carotid plaques. In addition to findings supporting the use of ^18^F-FDG as a marker of arterial inflammation, preliminary studies have pointed out the value of this tracer to predict plaque rupture and clinical events [18]. Despite the widespread use of ^18^F-FDG in cardiovascular disease, the non-specific uptake of the radiopharmaceutical is the most important drawback for imaging atherosclerosis [5]. To overcome these limitations, new probes have been studied that allow for the identification of vulnerable plaques with better specificity, with potential not only for localization but also for the identification of the cellular components and pathophysiological status of the plaque. Some of these molecules are ^18^F-methylcholine (FMCH), also uptaken by activated macrophages in vulnerable plaques [19]; ^68^Ga-DOTA-NOC-TATE-TOC, uptaken by somatostatin receptors expressed in activated lymphocytes and macrophages of the inflammed vulnerable plaque [20]; and ^18^F-sodium fluoride, whose uptake is associated with active calcification, apoptosis, and necrosis, with good specific characteristics for the detection of high-risk microcalcification and early unstable atherosclerotic disease [13].

Nevertheless, differences in histology and natural history occur between coronary plaques and large peripheral vessel plaques, such as those affecting carotid arteries. In coronary plaques, the calcium content seems to be an early and prognostic factor, whereas in large peripheral vessel plaques, the lympho-monocytic infiltration probably plays a major role in predicting vulnerability [21,22].

In coronary plaques, rupture is independent of plaque size, therefore there is the imperative need to differentiate stable from potentially unstable lesions in clinical practice [2]. The rationale for PET-based molecular imaging in this scenario is to better understand the pathogenic mechanisms underlying atherosclerosis to use the information to evaluate treatment efficacy as well as to identify those patients with the highest risk of events related to plaque rupture [13].

In this review, we will concentrate on the events occurring in coronary plaques and the role of PET/CT in imaging vulnerable coronary plaques. In particular, this review describes state-of-the-art imaging, with radiolabelled somatostatin analogues, and the processes associated with plaque rupture.

## 2. Molecular Events in Atherosclerosis

Atherosclerosis is liable for the majority of hard events and is characterized by the accumulation of lipids, inflammatory cells, and connective tissue within the arterial wall [23]. Although atherosclerotic plaques may remain quiescent for a long period, the disease can become life-threatening when plaque ruptures, leading to thrombus formation and occlusion of the vascular lumen [24]. Thus, the unnoted course of atherosclerosis makes it necessary to develop predictors of acute events long before symptomatic lesions develop. Rupture-prone plaques often do not cause significant obstruction [25] and may not be detectable by stress imaging. For this reason, early identification of the vulnerable plaque could add a prognostic value and has become a major focus in cardiovascular diagnosis [4].

Today, it is accepted that endothelial cell dysfunction is the earliest detectable physiologic manifestation of coronary atherosclerosis [26] and that smoking, high LDL levels, hypertension, and diabetes contribute to endothelial dysfunction. A normal endothelium has antithrombotic, anti-inflammatory, and vaso-modulatory functions which are canceled in atherosclerosis [27]; the consequence is an enhancement of the permeability to lipids and inflammatory cells (monocytes and T-lymphocytes) from the blood [28]. Together, the presence of endothelial dysfunction (increased adhesion of molecules and expression of inflammatory genes) and high circulating levels of atherogenic lipoproteins increases the deposition of lipid-laden, monocyte-derived foam cells in the subendothelial layer, conforming the atherosclerotic lesion (macrophages take up oxidized LDL deposited in the intima by different receptors, including scavenger receptor A (SR-A) and CD 36). Vascular smooth muscle cells migrate as a consequence of angiogenesis stimulation and extracellular matrix components are synthesized to build the fibrous cap, which contains inflammatory cells (macrophages and T-lymphocytes) [29]. Cellular and extracellular components are determinants for plaque stability, where the size of the lipid core, the bulk of the fibrous cap, and the paucity of smooth muscle cells impact the probability of disruption [30]. Inflammatory cells, particularly macrophages, break down matrix proteins in the fibrous cap through the secretion of metalloproteinases [31] and also secrete cytokines such as IFN-γ, interleukin 1B, and tumor necrosis factor B (TNFB), promoting the Th1 lymphocyte pathway which is strongly proinflammatory and induces apoptosis [32]. This pathway determines the conversion from stable to unstable plaque; in fact, inflammation favors the destruction of the fibrous cap and subsequent thrombosis. T-cells are also important for identifying autoantigens, such as oxidized LDL and heat-shock proteins, which function as targets for an autoimmune response that promotes cytokines’ release and stimulates inflammation as well as plaque formation.

## 3. Biology of Coronary Plaque

The atherosclerotic plaque develops as a consequence of an endothelial lesion and the entrance of LDL cholesterol into the intimal layer of the vessel wall. The consequence of this lesion is monocytes recruitment into the vessel wall and their differentiation into macrophages, which take up modified LDL [33]. The macrophages take up the oxidized LDL and trigger the inflammatory cascade, stimulating angiogenesis and the formation of intraplaque neo vessels, enabling the entrance of red blood cells, leukocytes, lipids, and oxidized lipoproteins [34]. Phagocytosis of red blood cells by macrophages leads to iron collection, which facilitates lipid peroxidation and further stimulates the recruiting of new macrophages as well as expansion of the plaque. In turn, macrophage apoptosis and the release of matrix metalloproteases (MMPs) destabilize the extracellular matrix; thereafter, micro-calcifications appear as part of the healing process. Plaque disruption takes place as a result of intra-plaque hemorrhage and necrosis, producing thinning, erosion, and disruption of the fibrous cap. Furthermore, in the arterial lumen, the contact between the plaque and blood facilitates thrombus formation, enhancing the risk of embolism (Figure 1) [35].

Macrophages deserve special mention because the outcome of the plaque is mainly determined by the actions of these cells, which are found in high densities in unstable coronary lesions [37], and their accumulation is proportional to atherosclerosis process [38]. Inflammation triggers the release of cytokines that promote macrophages into morphologically and functionally distinct phenotypes [39,40]. Foam cells present in the plaque core belong to the M1 lineage macrophages and their production is stimulated by IFN-γ secreted by CD4 lymphocytes [41].

## 4. The Vulnerable Plaque

Libby et al. proposed a broader hypothesis to explain the plaque rupture mechanism, which is an expansion of the original work described by Henney et al. [42]. These authors point out that cytokines become the critical factor in the inflammation cascade, driving the expression of proteases and counteracting the actions of proteolytic inhibitors. The hypothesis of Hansson et al. [43], deserves special attention suggesting that specific antigens elicit a T-cell response and that disease progression may be boosted by autoimmune responses to oxidized lipoproteins. It is well recognized that arterial thrombosis contributes to the onset of acute coronary syndromes and that the majority of deaths precipitated by acute thrombosis are associated with the rupture of an advanced plaque [44], which is critical in transforming asymptomatic atherosclerosis into an acute clinical event. The risk of plaque rupture is known to depend more on plaque composition than on the degree of vascular stenosis [45], being the most vulnerable those having a larger lipid core, an inflammed thin fibrous cap [46], lower collagen content, higher macrophage density, fewer smooth muscle cells, a larger lipid-rich necrotic core, spotty calcification, neovascularization, intraplaque hemorrhage, and expansive remodeling [13,47]. Furthermore, smooth muscle cells and macrophage apoptosis contribute to the enlargement of the necrotic core, increasing the risk of rupture [46,48].

In addition to the previously described events during plaque progression, calcium deposition also occurs by differentiation of mesenchymal cells into osteoclasts and osteoblasts. As a consequence of necrosis, inflammation and cell apoptosis, microcalcification appears as a repair mechanism [49].

## 5. Functional Imaging of Atherosclerosis

Since atherosclerosis is a systemic, mainly lipid-driven inflammatory disease leading to multifocal plaque development [50], metabolic images have emerged as a valid alternative to evaluate the risk of plaque rupture. PET radiopharmaceuticals that bind to components of the plaque or to the thrombus surface have allowed the identification of different mechanisms of plaque vulnerability, such as lesion components, inflammation, thrombosis, neoangiogenesis, hypoxia, apoptosis, microcalcifications, and the accumulation of matrix metalloproteinases [13].

These physiopathological events have been used as potential targets for molecular imaging agents labeled with radioisotopes. Radiopharmaceuticals available for targeting atherosclerotic plaques are listed in Table 1.

## 6. Peptides and Radiolabeled Somatostatin Receptors in Vulnerable Plaque Detection

In vivo visualization of inflammation at the cellular and molecular level has been possible through the use of different peptide receptor ligands and monoclonal antibodies. [90]. Overexpression of somatostatin (SST) receptors (SSTR) by inflammatory cells, immune cells, and blood vessels, among other factors, has made possible the use of several radiolabeled somatostatin analogues with different affinities for these receptors in a set of various inflammatory scenarios [91].

SST was first isolated from the ovine hypothalamus and characterized as a tetradecapeptide [92]. SST-producing cells are typically neurons or endocrine-like cells found in high densities in different organs and immune cells [93]. SST is a cyclic hormone that regulates diverse physiological cell processes via specific SSTR expressed by nerve cells, neuroendocrine cells, and inflammatory cells such as lymphocytes, monocytes, macrophages, peripheral blood mononuclear cells, and thymocytes [91]. Five SSTR subtypes have been described and found in cells involved in inflammatory responses with a high density expression as observed in neoangiogenic and peritumoral vessels, epithelioid cells, proliferating synovial vessels, and activated lymphocytes and monocytes [93]. These are G protein-coupled receptors located at the cell membrane which recognize the ligand and generate a trans-membrane signal. The distribution of these receptors varies in each tissue: SSTR subtypes 2 and 3 have been found in the peripheral blood, mononuclear cells and the spleen; SSTR subtypes 1, 2, and 3 in the thymus; SSTR subtype 2 in macrophages and dendritic cells; SSTR subtype 3 in B lymphocytes; and SSTR subtypes 1 to 5 in T lymphocytes [91], providing the molecular basis for many clinical applications of radiolabelled SST analogues [94]. Different SST analogues have been described for clinical practice; for example, pentetreotide labeled with ^111^In, with high affinities for type 2 and 5 SSTR, has been a popular radiotracer for imaging purposes. Today, new PET radiopharmaceuticals for somatostatin receptor scintigraphy are commercially available, showing better affinity for different receptors, such as ^68^Ga-DOTA-TATE and ^64^Cu-DOTA-TATE (affinity to type 2 and 5 receptors); ^68^Ga-DOTA-TOC (selective for 2 and 5-type receptors); and ^68^Ga-DOTA-NOC (affinity to 2, 3, and 5-type receptors) [95,96].

Monocytes and macrophages as triggers of vascular inflammation play an important role in the pathogenesis of atherosclerosis, contributing to necrotic core formation, fibrous cap thinning, and plaque vulnerability [97]. These cells express SSTR-2 on the surface in cultures [98] so that ^68^Ga-DOTA-TATE has the potential to become a surrogate marker of inflammation for the study of plaque biology [59]. Complex connections between the stage of plaque evolution (chronic vs. acute), macrophage density and activity, SSTR subtype expression, and ^18^F-FDG and ^68^Ga-DOTA-TATE uptake are known today. So far, evidence shows a tendency to confirm the utility of radiolabeled SSTR with ^68^Ga-DOTA-TOC to identify vulnerable plaques, with a good correlation with cardiovascular risk factors [99]. The added value of DOTA-TOC relies on its high specificity for detecting inflammation and plaque vulnerability compared to ^18^F-FDG, which gives only global information about the inflammatory burden [59].

^68^Ga-DOTA-octreotide tracers have been positioned as novel molecules for PET imaging due to physical characteristics, good spatial resolution, high sensitivity, and possibility for the quantitative assessment. The most used ^68^Ga-DOTA peptides are ^68^Ga-DOTA-TOC, ^68^Ga-DOTA-NOC, and ^68^Ga-DOTA-TATE. Recently, a new generation of radioligands based on SSTR-2 antagonists was described in a preclinical study, showing more favorable pharmacokinetics than agonists [100]. In this experience, the authors used ^111^In-DOTA-JR11, demonstrating favorable biodistribution and targeting efficiency for detecting atherosclerotic plaque, with a higher target-to-background ratio contrast than that observed with DOTA-TATE (agonistic radioligand). After incubating a human endarterectomy sample with ^111^In-DOTA-JR11, they demonstrated uptake of the radioligand in sites where immunohistochemistry showed SSTR- 2 receptors and noticed no radioligand uptake or SSTR 2 expression in areas of macrocalcifications which were visible in CT.

Pre-clinical studies have supported the use of ^68^Ga-DOTA-TATE for targeting activated macrophages in atherosclerotic plaques by detecting sites of SSTR-2 accumulation in the human body. Xiang et al. [101] assessed the binding of ^68^Ga-DOTA-TATE to macrophages and the normal endothelium ex vivo, as well as the uptake in inflammatory plaques in a mouse model. They reported the presence of SSTR-2 in activated macrophages and showed a high specific affinity of ^68^Ga-DOTA-TATE to atherosclerotic lesions by autoradiography. The potential of this probe in clinical settings has been confirmed by different authors.

As an interesting novel approach, Schatka et al. [102] reported that ^68^Ga-DOTA-TATE could provide a theranostic opportunity to characterize and modulate atherosclerotic plaque biology; in 11 oncological patients the baseline ^68^Ga-DOTA-TATE scans showed abnormal focal uptake in carotid vessels, which, after peptide receptor-targeted radionuclide therapy (PRRT), showed significant reduction in follow-up scans. They also documented a correlation between cardiovascular risk factors and radiotracer uptake at baseline; reported variables were: patient age (r = 0.76, *p* < 0.01); number of calcified plaques (r = 0.84, *p* < 0.001); and presence of hypercholesterolemia (vessel uptake was 71.6 +/−3.9 vs. 35 +/− 18.3 with no hypercholesterolemia; *p* = 0.004). They also noticed that PRRT-related reduction of ^68^Ga-DOTA-TATE uptake was stronger in non-calcified vs. calcified active plaques. Although this observation was focused on carotid plaques, it could serve as a starting point for further exploration of radionuclide-based anti-atherosclerotic molecular interventions aimed to decrease the degree of inflammation in high-risk atherosclerosis. In a SR Anzola et al. [99], analyzed the published data regarding the use of radiolabeled SSTRs for the assessment of endothelial inflammation in a total of 262 patients. Tarkin et al. [20] conducted a prospective study to validate the use of ^68^Ga-DOTA-TATE as a vascular inflammation imaging agent (VISION STUDY); they confirmed that high-target SSTR2 gene expression occurs exclusively among activated proinflammatory M1 macrophages in atherosclerosis and demonstrated the presence of SSTR-2 receptors from patients with coronary vascular disease. They found that ^68^Ga-DOTA-TATE correctly identified culprit coronary and carotid arteries in patients with acute coronary syndromes. Intra-observer correlation showed a coronary artery intraclass coefficient of 0.90, with an inter-observer correlation of 0.96. In culprit acute coronary syndrome lesions, ^68^Ga-DOTA-TATE uptake was higher than in the non-culprit segment (*p* = 0.008). According to ROC analysis, coronary ^68^Ga-DOTA-TATE mTarget to background ratio maximum values (TBRmax) greater than 2.66 showed 87.5% and 78.4% sensitivity and specificity, respectively. ^68^Ga-DOTA-TATE also identified high-risk stable lesions in low-attenuation plaques detected by CT, where mTBRmax > 2.12 showed a sensitivity and specificity of 83.3% and 71%, respectively. They also reported a significant correlation between clinical risk factors and ^68^Ga-DOTA-TATE uptake (age r = 0.44, *p* = 0.0004; total cholesterol r = 0.51, *p* < 0.0001; Framingham risk score r = 0.53, *p* < 0.0001). In a multivariate linear regression analysis, they demonstrated that these clinical factors were significant predictors of ^68^Ga-DOTA-TATE uptake. For patients receiving statins, they observed lower uptake providing anecdotal evidence that this probe may be used for monitoring the anti-inflammatory effects of specific drugs. The authors also compared the results with those obtained using ^18^F-FDG, and found a superiority of ^68^Ga-DOTA-TATE to discriminate high-risk vs. low-risk coronary atherosclerotic lesions, which was explained by the non-specific measurement of glucose metabolism of ^18^F-FDG in plaque cells (Figure 2).

Simon Wan et al. [60], in a descriptive study involving 20 patients with recent stenotic carotid events, did not find any significant ^68^Ga-DOTA-TOC uptake in the plaques, interpreted as absence of activated macrophages expressing SSTR2, and were not able to demonstrate the utility of ^68^Ga-DOTA-TOC for evaluating the inflammed plaque. The results could be explained by the presence of different phenotypes of macrophages in these particular lesions.

Malmberg et al. [64], in an observational study of 60 consecutive patients compared the diagnostic performance between ^68^Ga-DOTA-TOC and ^64^Cu-DOTA-TATE for detecting atherosclerotic plaques in different vessels, and observed superiority of ^64^Cu-DOTA-TATE explained by the better physical features of ^64^Cu. They reported a significant association between Framingham risk score and the overall maximum uptake of ^64^Cu-DOTA-TATE (r = 0.4, *p* = 0.004), suggesting a possible use of this radiotracer as a non-invasive biomarker of cardiovascular risk. Pedersen et al. [65], by using ^64^Cu-DOTA-TATE/MRI in a population of 10 patients with clinical symptoms of stroke or transient ischemic attack, reported a high uptake of the radiotracer in the lesions in symptomatic patients with a high correlation, with the presence of CD163 in activated macrophages at histology. Xian Li et al. [103], in a retrospective series of 16 patients, simultaneously compared ^18^F-FDG and ^68^Ga-DOTA-TATE PET/CT and reported significantly increased uptake in fibrotic and vulnerable plaques in large arteries compared to normal arteries. The comparison between the two molecules showed few differences in the low-risk group; nonetheless, in the high-risk group, ^68^Ga-DOTA-TATE showed significantly higher values than ^18^F-FDG. They also reported a significant correlation between the mean uptake, the clinical risk scores, and the presence of calcification. Mojtahedi et al. [61], by using ^68^Ga-DOTA-TATE PET/CT in a population of 44 patients with neuroendocrine tumors (NET), reported that the TBR value in the normal group was lower than in the group with atherosclerotic plaques (*p* < 0.0001) as well than in the group with fibrotic plaques (*p* = 0.0043) They found a significant correlation between ^68^Ga-DOTATATE uptake and the progression to the formation of atherosclerotic plaques based on the coronary CT calcium score (*p* = 0.0026). When they compared risk factors with ^68^Ga-DOTA-TATE TBR values, they found a direct correlation between the two variables (*p* = 0.0068). These findings support the potential of ^68^Ga-DOTA-TATE PET/CT for the molecular assessment of coronary artery disease. Rominger et al. [62], correlated ^68^Ga-DOTA-TATE uptake in the LAD coronary artery with the presence of calcified plaques and cardiovascular risk factors. Higher uptake allowed the distinction between patients with and without coronary calcifications (r = 0.34); ROC analysis showed that a TBR greater than or equal to 1.5 could differentiate between the two groups. The authors observed a significant correlation between the TBR values and prior vascular events (r = 0.26, *p* < 0.05), suggesting the potential of the molecule to detect the presence of high risk vulnerable plaques.

A report from Adams et al. [63] showed an overexpression of SSTR-2 when human umbilical vein endothelial cells are proliferating; this finding could probably support the active role of these receptors in angiogenesis, leading to a potential use of SSTRs for evaluating angiogenesis in the context of unstable plaques. Consequently, ^68^Ga-DOTA-TATE could be used to identify unstable plaque in two ways: the detection of activated macrophages and the angiogenesis within the atherosclerotic lesion [104].

Available evidence shows a tendency to support the utility of radiolabeled SSRS with ^68^Ga-DOTA-TOC to detect high risk, vulnerable, atherosclerotic plaques, and confirms a good correlation with risk factors, with the added value of high specificity for targeting inflammation and providing in vivo information on plaque vulnerability. Non-nuclear imaging modalities can give superb anatomic detail but are not able to look at early functional events and plaque vulnerability. The most important contribution of these radiotracers is their ability to detect inflammatory activity of the plaque, improving risk stratification and the prediction of hard clinical events. Furthermore, the information obtained could be potentially used for therapeutic decision-making and monitoring of anti-inflammatory treatment with statins. Nevertheless, SSTRs for PET imaging of coronary vulnerable plaques share the same technical limitations observed with other radiotracers used for these clinical indications, such as the limited spatial resolution, the size of the lesions with a low signal from the plaque and the partial-volume effect from adjacent structures. Other limitations are cardiac and respiratory motion artifacts entailing the need of specialized software to control these effects. Finally, while quantification is an important advantage of PET imaging, being the standardized uptake value (SUV) the most utilized parameter, an appropriate quantitative index for ^68^Ga-DOTA in atherosclerosis has not been established, which is important for an objective evaluation of the findings and to follow up in a clinical setting.

Regarding future perspectives, hybrid molecular images using advanced systems such as PET/MRI together with more precise quantitative analysis will enable the simultaneous characterization of morphologic and biologic features of the vulnerable plaque, providing a greater potential for tissue characterization and risk assessment. New PET radiotracers [5], such as ^18^F-aluminium, have been proposed for labelling biomarkers of vascular calcification such as matrix G1a protein (MGP), which is expressed by endothelial cells, fibroblasts, and chondrocytes. MGP binds to calcium crystals and may be suitable as a molecular imaging probe for the investigation of early vascular calcium deposits [105]. Another interesting potential biomarker of vascular calcification is bone morphogenetic-2 (BMP-2) that binds to calcium crystals [106]. Antibody fragments are also of great interest for imaging early stages of microcalcification.

Since inflammation starts with endothelial activation, the expression of selectins and vascular cell adhesion molecule-1 (VCAM-1) constitute potential imaging targets on the activated endothelium [10] as has been reported by Nakamura et al. and Nahrendorf in preclinical studies with promising results by using ^64^Cu-labelled monoclonal antibodies for P-selectin and ^18^F-labelled peptide compounds for VCAM-1 imaging [107]. Recently, Barret et al. [108] reported the results of a dual-isotope SPECT protocol for assessing the capability of ^111^In-Danbirt and ^99m^Tc-Demotate to determine atherosclerotic plaque inflammation and phenotypes. These authors reported that ^111^In-Danbirt discriminates the presence of different leukocyte subsets and differentiates the plaque phenotype of fibrous cap atheroma from both pathological intimal thickening and fibrocalcific plaque segments. Detecting the presence of leukocyte subsets of proinflammatory macrophages may improve the understanding of inflammation involvement in atherosclerotic progression.

## 7. Conclusions

Atherosclerosis is a systemic and multifocal disease of inflammatory origin with a silent course that makes it necessary to develop predictors of disease before symptomatic lesions develop. To date, new imaging tools have been studied in different preclinical and clinical scenarios which have provided knowledge about the presence, extension, and progression of the disease, and about the vulnerability of atherosclerotic lesions. To have the possibility to identify the atherosclerotic plaque in terms of molecular and cellular processes, focusing the aim of molecular imaging is crucial, which is the development of a tailored imaging radiotracers to characterize vulnerable plaque, leading to a better outcome through risk stratification, with the perspective to customize pharmacological treatment and optimize results. The ability of ^68^Ga-DOTA-TATE/TOC/NOC to provide prognostic information and evidence of treatment efficacy has been demonstrated; however, large-scale trials are required to determine the incremental value of plaque imaging in risk assessment and its influence on clinical outcome.

## Figures and Tables

**Figure 1 jcm-10-05515-f001:**
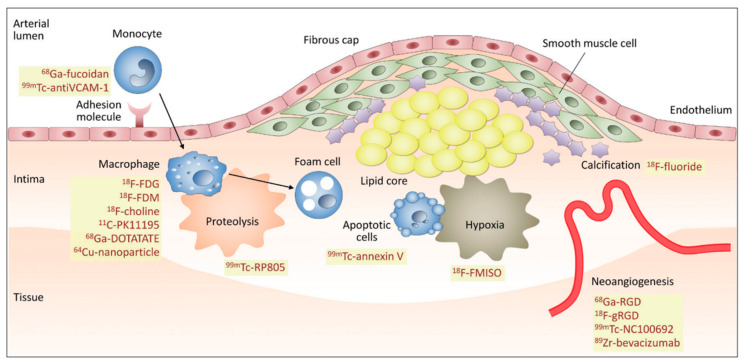
Pathogenesis mechanism and molecular imaging targets in vulnerable plaque. Schematic representation of event cascade leading to plaque inception and destabilization, and specific radiotracers to detect different events (inflammation by activated macrophages, apoptosis, neoangiogenesis, and calcification). Reprinted with permission from ref. [36].

**Figure 2 jcm-10-05515-f002:**
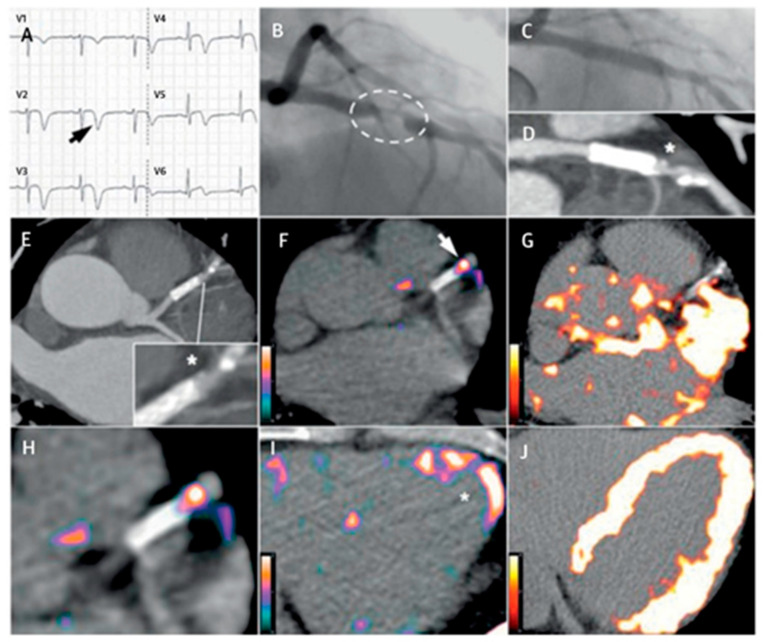
Comparison between ^68^Ga-DOTATATE and ^18^F FDG. Culprit left anterior descending (LAD) coronary artery stenosis in a 57-year-old man with acute coronary syndrome. (**A**) ECG showing ST-T segment elevation and inverted deep T-waves in V2-V5. (**B**) Angiography detected severe obstructive LAD lesion. After percutaneous coronary stenting, residual coronary plaque is seen with low attenuation and spotty calcification (**C**–**E**; star symbol depicts location of residual lesion). ^68^Ga-DOTA-TATE PET shows intense focal uptake, consistent with inflammation of high-risk atherosclerotic plaques in the distal portion of the stented culprit lesion (**F**,**H**,**I**). ^18^F-FDG PET (**G**,**J**) shows myocardial spillover, completely obscuring the coronary arteries. Adapted with permission from ref. [20].

**Table 1 jcm-10-05515-t001:** Radiopharmaceuticals used for imaging atherosclerotic plaques.

Target Process and Mechanism of Uptake	Imaging Target	Radiopharmaceuticals	References
A. Inflammation Targeting receptors in activated macrophages	Glucose metabolism in macrophages	^18^F-FDG	[51,52]
	Expression of the TSPO 18KDa translocator protein	^11^C-PK11195	[53,54]
		^18^F-FEDAA1106	[55]
		^18^F-FEMPA	[56]
		^18^F GE-180	[57]
	Microcalcification cytokines derived from macrophages (IL6- IFN-γ) increase vascular smooth muscle cell calcification	^18^F-NaF	[58]
	SSTR-2 Somatostatin receptor2 expressed on activated macrophages	^68^Ga-DOTA-TATE/NOC/TOC	[18,57,59,60,61,62,63]
		^64^Cu-DOTA-TATE	[64,65]
	Chemokine receptor 4 expressed in macrophages and CD68 involved in atherogenic process	^68^Ga-Pentixafor	[66]
	FR Beta localize in the folate receptor Beta expressed on the surface of activated macrophages	^18^F-FOL	[67]
	Targets COX2 which catalyze the production of PGE2	^11^C-PS13	[68]
		^11^C-MC1	
	Choline uptake in activated macrophages	^18^F-FCH	[69,70]
B. Angiogenesis	AlfaVBeta3 integrins that play important roles in foam-formation cells	^18^F Galacto-RGD	[71]
		^68^Ga-NOTARGD	[72]
		^18^F-Fluciclatide	[73,74]
	Alfa7nAchR Nicotinic receptor whose activation increases the LDL cholesterol uptake within macrophages	^18^F-ASEM	[75,76]
		^11^C-NS14492	
		^18^F-FNS14490	
C. Hemorrhage	Glycoprotein IIB/IIIA complex expressed in activated platelets	^18^F-GP1	[77]
	Fibrins	^99m^Tc fibrin-alfa chain peptide	[78]
D. Hipoxia	Tracer becomes reduced and trapped within hypoxic cells	^18^F-FMISO	[79,80,81]
		^18^F-HX4	[82]
E. Apoptosis	Tracer binds to cell membranes within apoptotic cells	^18^F-ML10	[83]
		^18^F-Anexin V	[84,85]
F. Extracellular Matrix	Matrix metalloproteinases released by activated macrophages	^18^F/124I	[5]
G. Atherosclerotic Lesion Components	Endothelium	P-Selectin V-Cam1	[86,87]
	Foam cells	^99m^Tc-LOX-1mAb	[88]
	Lipids	^99m^Tc-MDA2	[89]

## Data Availability

Not applicable.

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
