# Peer review of "Molecular Imaging of Vulnerable Coronary Plaque with Radiolabeled Somatostatin Receptors (SSTR)"

_jcm, 2021, doi:10.3390/jcm10235515_

Round 1
Reviewer 1 Report
The manuscript is well written and analyzes the pathophysiology of vulnerable atherosclerotic plaque. There is also a detailed review of trials searching for molecular biomarkers detected in vulnerable plaques by PET scan.
1. In my opinion the structure of the review is fine and the context includes a contemporary analysis of literature.
However the authors should refer more extensively to the limitations of the methods and to the potent clinical impact of the presented data.
2. Are these methods compatible with the analysis of Atheroslerotic plaque by CT coronography?
3. Please define a small typo in line 172
Thank you in advance.
"T, which is critical..." ???"
L
Author Response
Manuscript: JCM-1444191
Title: MOLECULAR IMAGING OF VULNERABLE CORONARY PLAQUE WITH RADIOLABELED SOMATOSTATIN RECEPTORS (SSTR).
Special Issue: New Horizons and current concepts in cardiac computed tomography.
Dear Editor Ms Ann Wang
We appreciate the opportunity to resubmit a revised version of our manuscript. We thank the reviewers for their comments that will certainly improve the quality of the paper and we provide below a point-by-point reply on the issues that were raised by the reviewers. We also submit a revised version of the manuscript based on the reviewers’ comments, with track changes.
Reviewers' comments:
Reviewer 1
We thank the Reviewer 1 for his/her encouraging words and we now provide our responses to the comments.
- In my opinion the structure of the review is fine and the context includes a contemporary analysis of literature.
- However the authors should refer more extensively to the limitations of the methods and to the potent clinical impact of the presented data.
R: We have expanded the paragraph that concludes the review of SSTRs in vulnerable plaque, emphasizing on the clinical impact and the limitations of the technique, as follows: (Lines 383-397)
Evidence so far, show a tendency to confirm the utility of radiolabeled SSRS with 68Ga-DOTA-TOC to detect high risk, vulnerable atherosclerotic plaques and an interesting correlation with risk factors, with the added value of high specificity for targeting inflammatory changes to provide in vivo information on the vulnerability of plaques. Non-nuclear imaging modalities present a lot of possibilities, but are not able to look at early events and plaque vulnerability. The most important contribution of these radiotracers is to detect the inflammatory activity of the plaque, improving risk stratification and the prediction of hard clinical events. Furthermore, the information obtained from the results could be potentially used for therapeutic decision-making and monitoring of anti-inflammatory treatment with statins. Nevertheless, SSTRs for PET imaging of coronary vulnerable plaques share the same technical limitations observed with other radiotracers for the same indication, being the most important factor the limited spatial resolution and the size of the lesions to be detected, with a low signal from the plaque and partial-volume effect from adjacent structures. Other limitations are cardiac and respiratory motion artifacts entailing the use of specialized software to control these effects. Finally, while quantification is an important advantage of PET imaging, being the standardized uptake value (SUV) the most utilized parameter, an appropriate quantitative index for 68Ga-DOTA in atherosclerosis has not been established. This would be important for objective evaluation of the findings and to follow up in a clinical setting.
- Are these methods compatible with the analysis of Atheroslerotic plaque by CT coronography?
R: We have added a paragraph in the introduction allusive to the synergistic role of both imaging modalities.
Because the dynamic nature of coronary atherosclerotic lesions, it is not easy to identify which plaques will become unstable, based just on a single imaging modality. An approach that allows to identify the cap thickness, the necrotic core areas and the degree of inflammation of the plaque is of paramount clinical relevance. Given the involved costs, the time required and the potential adverse effects, invasive techniques are not suitable for screening asymptomatic patients. CT angiography can be used to identify the presence of lesions with high risk characteristics, i.e, those with areas of low attenuation and expansive remodeling. Serial CT could also determine dynamic changes in the plaque as an indicator of instability, warning the clinician to adopt prophylactic measures. In this scenario, it is also important to consider the group of asymptomatic patients with small, angiographically invisible plaques with a potential to rupture due to inflammatory changes that could precipitate a hard event; in such cases, metabolic images puts in evidence the synergistic role of both imaging modalities.
- Please define a small typo in line 172
R/:Typo T was a mistake and was deleted

Reviewer 2 Report
Overall, the review of the molecular imaging biomarkers for atherosclerosis was very well done and an interesting read. It it clear from this article that there are some promising biomarkers that may add substantial clinical value for the purpose of earlier detection and risk stratification of atherosclerotic lesions.
Although, all of my comments are minor, the largest concern is with grammar and use of the English language.
- Line 95 - In many places there is no space between the word and the citation (i.e. practice[2]. In other areas of the manuscript there was a space between the word and citation. Please put a space between the citation and the word preceding it consistently throughout the manuscript.
- Line 116 - misspelled word (used hast, likely meant has)
- Line 122 - it reads as A(SR-A and CD36). I presume the A before the parentheses is a mistake?
- Line 172 - The start of the sentence says "T, which is .....; what is T? Were you referring to T lymphocytes or a particular T cell type?
- Line 237 - What is SRS? I did not see this defined earlier in the manuscript. SST and SSTR were defined but could not find a definition for SRS. Please define
- Line 257 - after the citation, you need a comma
- Line 268 - reword your sentence, it is hard to follow
- Line 340 - You used the words interesting correlations which was also used earlier in the manuscript. Please expand on those correlations, what was the uptake of the radiolabel correlated with and what was the interpretation of those correlates
- Figure legends need to provide more information to help the reader understand the figure. For example, in figure 2, there are arrows and anatomy is circled in some of the panels and yet there is no information telling the reader what tracer they are looking at in which panels and what it tells us. Please provide more detailed explanation in your figure legends.
- Lastly, could you comment on the size of the atherosclerotic plaques as this may be necessary given that for most clinical PET scanners, spatial resolution is usually around 4-5mm at best. If lesions are smaller than this, how applicable are PET tracers when each image voxel size might be larger than an actual atherosclerotic lesion. There was a mention at the beginning about spatial resolution issues with PET, but nothing thereafter. Please expand on that conversation or perhaps add a paragraph regarding the current limitations to molecular imaging in this field and how current improvements can eradicate this issue.
Author Response
Manuscript: JCM-1444191
Title: MOLECULAR IMAGING OF VULNERABLE CORONARY PLAQUE WITH RADIOLABELED SOMATOSTATIN RECEPTORS (SSTR).
Special Issue: New Horizons and current concepts in cardiac computed tomography.
Dear Editor Ms Ann Wang
We appreciate the opportunity to resubmit a revised version of our manuscript. We thank the reviewers for their comments that will certainly improve the quality of the paper and we provide below a point-by-point reply on the issues that were raised by the reviewers. We also submit a revised version of the manuscript based on the reviewers’ comments, with track changes.
Reviewers' comments:
Reviewer 2
We thank the Reviewer 2 for his/her encouraging words and we now provide our responses to the comments.
Overall, the review of the molecular imaging biomarkers for atherosclerosis was very well done and an interesting read. It it clear from this article that there are some promising biomarkers that may add substantial clinical value for the purpose of earlier detection and risk stratification of atherosclerotic lesions.
Although, all of my comments are minor, the largest concern is with grammar and use of the English language.
- Line 95 - In many places there is no space between the word and the citation (i.e. practice[2]. In other areas of the manuscript there was a space between the word and citation. Please put a space between the citation and the word preceding it consistently throughout the manuscript.
R/: The whole manuscript has been checked and the citation format is being corrected.
- Line 116 - misspelled word (used hast, likely meant has)
R/: misspelled word is being corrected.
- Line 122 - it reads as A(SR-A and CD36). I presume the A before the parentheses is a mistake?
R/: The paragraph has been edited to make clear the idea:
The combination of endothelial dysfunction (increased adhesion of molecules and expression of inflammatory genes) and high circulating levels of atherogenic lipoproteins leads to the accumulation of lipid-laden, monocyte-derived foam cells in the subendothelial layer, forming the early atherosclerotic lesion (macrophages take up oxidized LDL deposited in the intima by different receptors, including scavenger receptor A (SR-A) and CD 36 ). (lines 121 -122)
- Line 172 - The start of the sentence says "T, which is .....; what is T? Were you referring to T lymphocytes or a particular T cell type?
R/: Typo L was a mistake and was deleted.
- Line 237 - What is SRS? I did not see this defined earlier in the manuscript. SST and SSTR were defined but could not find a definition for SRS. Please define.
R/: SRS was a spelling mistake and it was checked and corrected by SST or SSTR accordingly within the manuscript.
- Line 257 - after the citation, you need a comma.
R/: The citation format has been checked within the manuscript.
- Line 268 - reword your sentence, it is hard to follow
R/: We have edited the paragraph as follows:
As an interesting novel approach, Schatka et al [100], reported how 68Ga-DOTA-TATE could provide a theranostic opportunity to characterize and modulate atherosclerotic plaque biology; these authors observed in 11 oncological patients, that the baseline 68Ga-DOTA-TATE scans showed abnormal focal uptake in carotid vessel walls which, after peptide receptor-targeted radionuclide therapy (PRRT), showed significant uptake reduction in follow up scans; (lines 232-234)
- Line 340 - You used the words interesting correlations which was also used earlier in the manuscript. Please expand on those correlations, what was the uptake of the radiolabel correlated with and what was the interpretation of those correlates.
R/: Following your suggestion we decided to expand by authors, the description of the different correlations between the variables and the 68Ga-DOTA quantitative analysis, as follows:
Schatka et al [101] also documented a correlation between cardiovascular risk factors and radiotracer uptake at baseline: patient age (*r=0.76, p<0.01); number of calcified plaques (r=0.84, p<0.001); presence of hypercholesterolemia (vessel uptake was 71.6+- 3.9 vs 35 +- 18.3 with no hypercholesterolemia, p=0.004). They also noticed that PRRT-related reduction of 68Ga-DOTA-TATE uptake was stronger in non-calcified versus calcified active plaques.
Tarkin et al. [21], - they confirmed that high target SSTR2 gene expression occurs exclusively among activated proinflammatory M1 macrophages in atherosclerosis and demonstrated the presence of SSTR-2 receptors from patients with coronary vascular disease; -they found that 68Ga-DOTA-TATE correctly identified culprit coronary and carotid arteries in patients with acute coronary syndrome; - the reproducibility of 68Ga-DOTA-TATE for intraobserver correlation showed a coronary artery intraclass coefficient of 0.90 and for interobserver correlation of 0.96. In culprit acute coronary syndrome lesions 68Ga-DOTA-TATE uptake was higher than the non- culprit segment (*p=0.008). According to ROC analysis, coronary 68Ga-DOTA-TATE mTBRmax values greater than 2.66 showed 87.5% and 78.4% sensitivity and specificity respectively; 68Ga-DOTA-TATE identified high-risk stable lesions in low attenuation plaques detected by CT; an uptake with mTBRmax > 2,12 showed sensitivity and specificity of 83.3% and 71%, respectively. They also reported significant correlation between clinical risk factors for coronary vascular disease and 68Ga-DOTA-TATE uptake (age r=0.44 p=0.0004; total cholesterol r=0.51 p< 0.0001; Framingham risk score r=0.53 p<0.0001). In a multivariate linear regression analysis, they demonstrated that these clinical factors were significant predictors of 68Ga-DOTA-TATE uptake;
Malmberg et al. [65], They reported a significant association between Framingham risk score and the overall maximum uptake of 64Cu-DOTA-TATE (r=0.4 p=0.004), suggesting the potential use of this radiotracer as a non-invasive biomarker of cardiovascular risk.
Mojtahedi et al. [61], by using 68Ga-DOTA-TATE PET/CT in a population of 44 patients with neuroendocrine tumors (NET) reported that the TBR value in the normal group was lower than in a group with atherosclerotic plaques (p<0.0001) and in the fibrotic plaque group ( p=0.0043). They found a significant correlation between 68Ga-DOTATATE uptake and the progression to formation of atherosclerotic plaques, based on coronary CT calcium score (p=0.0026). When they compared risk factors with 68Ga-DOTA-TATE TBR values, they found a direct correlation between the two variables(p=0.0068);
Rominger et al. [62] correlated the 68Ga-DOTA-TATE uptake in the left anterior descending coronary artery with the presence of calcified plaques and cardiovascular risk factors. Higher uptake allowed to distinguish between patients with and without coronary calcifications (r=0.34); the ROC analysis showed that a TBR greater than or equal to 1.5 could differentiate between patients with and without coronary calcifications. The authors observed significant correlation between the TBR values and prior vascular events, (r=0.26 p<0.05) suggesting the potential of the molecule to detect the presence of high risk vulnerable plaques.
- Figure legends need to provide more information to help the reader understand the figure. For example, in figure 2, there are arrows and anatomy is circled in some of the panels and yet there is no information telling the reader what tracer they are looking at in which panels and what it tells us. Please provide more detailed explanation in your figure legends.
R/: Figure legends have been improved for better understanding:
Figure 1. Pathogenesis mechanism and molecular imaging targets in vulnerable plaque. Schematic representation of event cascade leading to plaque inception and destabilization and specific radiotracers to detect different events (inflammation by activated macrophages, apoptosis, neoangiogenesis, calcification). Reprinted from Nuclear Molecular Imaging for Vulnerable Atherosclerotic Plaques SJ Lee, JC Paeng JC.,2015, Korean journal of radiology, 16 (5), 955–966.
Figure 2. Comparison Between 68Ga-DOTATATE (F,H,I) and 18F FDG (G,J). Coronary PET Inflammation Imaging. Culprit left anterior descending artery stenosis in a 57-year old man with acute coronary syndrome (B); after percutaneous coronary stenting, residual coronary plaque is seen with low attenuation and spotty calcification (D,E); 68Ga-DOTA-TATE PET shows intense focal uptake depicting inflammation of high risk atherosclerotic plaque in the distal portion of the stented culprit lesion (F,H,I); 18F-FDG PET (G,J) shows how myocardial spillover completely obscures the coronary arteries. Adapted from Detection of Atherosclerotic Inflammation by 68Ga-DOTATATE PET Compared to 18F FDG PET Imaging. JM Tarkin and FR Joshi, 2017, J Am Coll Cardiol, 11; 69.
- Lastly, could you comment on the size of the atherosclerotic plaques as this may be necessary given that for most clinical PET scanners, spatial resolution is usually around 4-5mm at best. If lesions are smaller than this, how applicable are PET tracers when each image voxel size might be larger than an actual atherosclerotic lesion. There was a mention at the beginning about spatial resolution issues with PET, but nothing thereafter. Please expand on that conversation or perhaps add a paragraph regarding the current limitations to molecular imaging in this field and how current improvements can eradicate this issue.
R/: We have added to the introduction a paragraph allusive to your important comment:
However, limited spatial resolution becomes the most important weakness for evaluating atherosclerotic plaques; for overcoming this limitation, hybrid systems have emerged as state of the art technology for simultaneous morphological and functional imaging [11,12]. Co-registration of PET images with CT or MRI gives a precise anatomical localization of the PET signal, and since uptake quantification by SUV may not reflect the true tracer uptake within the lesion, the concomitant use of contrast-enhanced CT or MRI could help to mitigate this source of error. Co-registered CT images permits attenuation correction, which is critical in the diagnostic approach of small structures; co-registered MRI images provide soft tissue contrast information and the possibility to evaluate the presence of thrombus. Because the quality of PET images could be affected by cardiac and respiratory motion, different technical strategies should be applied, such as the use of end-diastolic images [13] and software-based motion correction techniques like motion tracking with 4D CT and data-drive correction respiratory gating from 4D PET [4,5], improving the detection of small foci of uptake within the coronary arteries. High-resolution instruments have been recently introduced with the possibility to image lesions as small as 1-2 mm in size, like the new CZT cameras [14], and the total body digital PET [15]. Finally, as part of the strategies to improve the detection of small foci of uptake in the atherosclerotic plaques, the development of target-specific radiotracers has resolved the issue of interfering myocardial uptake and the necessity of faster clearance of the tracer from the blood stream.
